# Synergistic Amylase and Debranching Enzyme Catalysis to Improve the Stability of Oat Milk

**DOI:** 10.3390/foods14071271

**Published:** 2025-04-05

**Authors:** Xinyan Zhan, Jinye Zhang, Jiali Xing, Jinyi Xu, Dan Ouyang, Li Wang, Ying Wan, Xiaohu Luo

**Affiliations:** 1Zhejiang-Malaysia Joint Research Laboratory for Agricultural Product Processing and Nutrition, College of Food Science and Engineering, Ningbo University, Ningbo 315832, China; 2211390020@nbu.edu.cn (X.Z.); jinyezhang2023@163.com (J.Z.); xjy13155561316@163.com (J.X.); ouyangdan@nbu.edu.cn (D.O.); 2Key Laboratory of Detection and Risk Prevention of Key Hazardous Materials in Food, China General Chamber of Commerce, Ningbo Key Laboratory of Detection, Control, and Early Warning of Key Hazardous Materials in Food, Ningbo Academy of Product and Food Quality Inspection (Ningbo Fibre Inspection Institute), Ningbo 315048, China; hellojiali77@gmail.com; 3State Key Laboratory of Food Science and Technology, School of Food Science and Technology, National Engineering Research Center for Functional Food, Jiangnan University, 1800 Lihu Avenue, Wuxi 214122, China; wangli@jiangnan.edu.cn

**Keywords:** oat starch, oat milk, debranching enzyme, stability, enzymatic hydrolysis

## Abstract

Oat starch plays a crucial role in the stability of oat milk. Enzyme-hydrolyzed oat starch has been demonstrated to be an effective means of improving the stability of oat milk. The effects of different enzyme combinations on the stability of oat milk and the properties of starch in oats were investigated by adding α-amylase, amyloglucosidase, and different ratios of pullulanase and isoamylase. The results showed that as the degree of hydrolysis increased, the molecular weight, amylose content, and side chain length distribution of the starch decreased significantly. Moreover, compared with oat starch, the rheological and emulsifying properties of the starch hydrolysates were improved, and the characterization of emulsion stability showed that a 1:2 ratio of pullulanase to isoamylase promoted effective debranching and thus improved the stability of oat milk. This study demonstrated that debranching enzymes enhance the enzymatic hydrolysis of beverages and improve the physicochemical properties and stability of oat milk.

## 1. Introduction

Consumer demand for plant-based foods is increasing as a result of the growing global population, and the development of nutritionally balanced plant-based beverages is important in addressing the environmental challenges of population growth [1,2]. Among various plant-based alternatives, the oat ranks as one of the most significant cereal grains due to its well-balanced nutritional value and its richness in dietary fiber, notably β-glucan, a type of soluble dietary fiber that has been associated with reducing the risk of cardiovascular diseases, as well as phenolic compounds with antioxidant potential [3,4,5]. Moreover, oat milk, an emerging plant-based milk substitute, has gained popularity in recent years owing to the rising incidence of lactose intolerance among the general public, with the global oat milk market size reaching USD 644 million by 2023 [6].

Current oat-milk products still exhibit several shortcomings, including solid–liquid separation, floating fat, and poor flavor [7]. These issues negatively impact the quality of the product and limit its potential for further development. As the main component of oats, starch makes up nearly 60% of the dry matter in oat grains, and its insolubility is a major cause of instability in oat milk [8]. Suitable hydrolysis of starch during processing is a good way to solve the instability caused by starch and to improve the quality of the beverage. Previous studies have reported that by optimizing the slurry concentration, liquefaction time, and α-amylase concentration, the yield of oat milk can be increased from 53.92% to 78.87% [9]. Enzymatic extrusion by α-amylase resulted in the significant degradation of starch and contributed to the stability of oat milk [10]. In addition, the deep liquefaction and saccharification of oat core flour were utilized to increase the solubility of oat core flour and enhance the stability of oat milk [11]. Nevertheless, the requirements of enzymatic hydrolysis cannot be satisfied by the liquefaction and saccharification of starch alone.

As the enzymatic process gains prominence in the food industry, there will be a corresponding increase in the scrutiny of its efficiency. α-amylase is unable to hydrolyze the α-1,6-glycosidic linkage in starch and has a low bioconversion rate [12]. In contrast, debranching enzymes can boost production efficiency while imparting novel properties and functions to modified starch [13]. In the past few years, debranching enzymes have been utilized to uncover the correlations among structure, physicochemical properties, and functionality. It has been reported that the action of debranching enzymes on starch ultimately improves the thermal stability of resistant starch by generating a high concentration of short chains, lowering the molecular weight, and improving the formation of linear dextrins [14,15,16]. Hydrolysis of the α-1,6-glycosidic bond by the simultaneous use of α-amylase and isoamylase resulted in the formation of a special structure of starch, which improved the conversion of starch to maltose and showed the efficient debranching activity of isoamylase [17]. Debranching with pullulanase resulted in starch that was more crystalline and more irregular in shape, and the ratio of resistant to slow-digestible starch was altered after debranching [18]. Although debranching enzymes have been utilized in starch structure modification, the impact of the synergistic action between amylase and debranching enzymes on beverage stability remains unstudied. This study innovatively revealed the regulatory mechanism of the synergistic action between amylase and debranching enzymes on the stability of oat milk. Unlike the traditional single-enzyme modification mode, it utilized the synergistic action of multiple enzymes, providing a new technical pathway for the regulation of the stability of plant-based beverages.

This paper aimed to explore the effect of amylase and debranching enzyme synergy catalysis on the stability of oat milk. The degree of hydrolysis, particle size, and zeta potential of oat starch hydrolysates were systematically evaluated. In addition, the mechanism underlying the stabilization of starch in oat milk was explored by analyzing the fine structural alterations of the starch during enzymatic hydrolysis, including monosaccharide composition, molecular weight distribution, and chain length distribution. Last but not least, the rheological and emulsifying properties of the hydrolysates were studied to provide a theoretical basis for improving oat milk stability.

## 2. Materials and Methods

### 2.1. Materials

Oats were purchased from Wuchuan County Hechuan Green Food Co., Ltd. (Hohhot, China). α-amylase (2000 U/mL) and amyloglucosidase (50,000 U/mL) were bought from Henan Wanbang Chemical Technology Co., Ltd. (Zhengzhou, China). Pullulanase (100,000 U/mL) was purchased from Zhejiang Tianhe Food Biotechnology Co., Ltd. (Lishui, China). Isoamylase (100,000 U/mL) was bought from Shanghai Xingye Biological Science and Technology Co., Ltd. (Shanghai, China). Isoamylase (500 U/mL) was from Megazyme (Wicklow, Ireland). Rapeseed oil was provided by the Luhua Group (Yantai, China). All other chemicals (analytical grade) were purchased from local suppliers.

### 2.2. Single-Factor Tests of Enzyme Ratios

#### 2.2.1. Single-Factor Test of α-Amylase Addition

The preparation process for oat milk was modified according to the method of Ren et al. [10]. Oat slurry was obtained by grinding a 1:8 (*w*/*v*) mixture of oats and distilled water in a DEASS-PZ grinder (Shenzhen Wande Electric Appliance Co., Ltd., Shenzhen, China). To investigate the effect of α-amylase on oat milk stability, the oat slurry was treated with varying α-amylase concentrations (0.60%, 0.80%, 1.00%, and 1.20%, *w*/*w*, starch dry basis), along with 0.20% amyloglucosidase and 0.22% debranching enzyme (pullulanase: isoamylase, 1:1). The enzymatic hydrolysis was carried out for 60 min at 60 °C, followed by enzyme inactivation in 95 °C boiling water for 30 min. The mixture was filtered through a 100-mesh sieve to obtain oat slurry. Oat milk was manufactured by mixing oat slurry with 3% (*w*/*v*) rapeseed oil and then homogenizing the mixture with a T18 homogenizer (IKA, Staufen, Germany) for 2 min at 12,000 rpm.

#### 2.2.2. Single-Factor Test of Amyloglucosidase Addition

The effects of amyloglucosidase addition on the stability of oat milk were investigated by treating the slurry with varying amyloglucosidase concentrations (0.1%, 0.15%, 0.2%, 0.25%, and 0.3%), along with 1.00% α-amylase and 0.22% debranching enzyme (pullulanase: isoamylase, 1:1). Subsequent steps were identical to those described above.

#### 2.2.3. Single-Factor Test of Debranching Enzyme Addition

The effects of debranching enzyme addition on the stability of oat milk were investigated by treating the slurry with varying debranching enzyme concentrations (0.15%, 0.20%, 0.25%, 0.30%, and 0.35%, pullulanase: isoamylase, 1:1), along with 1.00% of α-amylase and 0.20% of amyloglucosidase. Subsequent steps were identical to those described above.

#### 2.2.4. Single-Factor Test of Different Proportions of Pullulanase and Isoamylase

The effects of different proportions of pullulanase and isoamylase on the stability of oat milk were investigated by treating the slurry with varying ratios of pullulanase and isoamylase (1:0, 2:1, 1:1, 1:2, and 0:1), along with 1.0% of α-amylase and 0.2% of amyloglucosidase. Subsequent steps were identical to those described above.

### 2.3. Stability Coefficient of Oat Milk

The stability coefficient of oat milk was determined according to the method of Su et al. [19], with modification. 50 mL of oat milk was poured into a centrifuge tube, then centrifuged at 450× *g* for 5 min. The supernatant was collected and diluted 100 times. The absorbance was measured using a T6 UV-Vis spectrometer (PERSEE, Beijing, China). The stability coefficient was determined according to Formula (1).(1)$$\mathrm{Stability}\;(\%)=A_{2}/A_{1}\times 100\%$$
where *A*_2_ is recorded as the absorbance of the supernatant diluted 100 times after centrifugation, and *A*_1_ is the absorbance of the supernatant diluted 100 times before centrifugation.

### 2.4. Extraction of Oat Starch (OS)

The method of extracting OS was according to Kaur et al. [20], with slight modification. Oat flour was mixed with petroleum ether at 1:2 (*w*/*v*) and stirred overnight to degrease. After centrifuging at 250× *g* for 5 min to remove the petroleum ether, the precipitate was dried at room temperature to obtain degreased oat flour. The resulting precipitate was mixed with NaOH (0.05 M) at 1:5 (*w*/*v*), stirred for 30 min. The mixture underwent centrifugation at 1350× *g* for 15 min to separate, and the supernatant was discarded. Afterwards, the precipitate was dispersed with distilled water and sieved through a 100-mesh sieve. The pH was adjusted to 7.0, and the precipitate was collected via another centrifugation process. Finally, the obtained precipitate was washed multiple times with distilled water and subsequently freeze-dried for 48 h. The OS was milled and sieved through a 100-mesh sieve for further use.

### 2.5. Preparation of Enzyme-Hydrolyzed Oat Starch (EHOS)

A 7.875% (*w*/*v*) oat starch suspension was prepared using 0.1 M sodium acetate buffer solution (pH 5.0) with magnetic stirring for 1 h. The sample suspension was placed in a boiling water bath and heated for 30 min, stirring every 10 min. After cooling to 60 °C, debranching enzyme was added to the oat starch (OS) paste at varying ratios (2:1, 1:1, and 1:2, *w*/*w*, starch dry basis) according to the optimal enzymatic process. Enzymatic hydrolysis was then conducted for respective durations (2, 25, and 40 min). Then, the temperature was adjusted to 95 °C to inactivate the enzyme. A portion of the enzymatic hydrolysates was collected for Z-average size, zeta potential, and rheological measurements, and another portion was freeze-dried for 48 h and ground to sieve through a 100-mesh sieve for further analysis. The samples with different enzymatic hydrolysis times and different debranching enzyme ratios were named as 2PI_2:1_, 2PI_1:1_, 2PI_1:2_, 25PI_2:1_, 25PI_1:1_, 25PI_1:2_, 40PI_2:1_, 40PI_1:1_ and 40PI_1:2_, respectively.

### 2.6. Characterization of Enzyme-Hydrolyzed Oat Milk, OS, and EHOS

The Z-average size and zeta potential of OS and EHOS were measured using a Zeta Sizer (Malvern Instrument Ltd., Malvern, UK). The enzymatic hydrolysis solutions were diluted 10 times with distilled water before measurement. The amylose content was determined using an amylose content assay kit (Solarbio, Beijing, China).

Reducing sugar content was measured at different time points (0, 0.2, 0.5, 1, 5, 10, 20, 30, 40, 50, and 60 min) taken during enzymatic hydrolysis, and the samples’ degrees of hydrolysis (*DH*) were calculated. Reducing sugar content was measured using a glucose assay kit (Beyotime, Shanghai, China). The degree of hydrolysis of oat milk, OS, and EHOS were determined according to Formula (2).(2)DH (%)=(G×V)/m×100%
where *G* is the content of reducing sugar(mg/mL), *V* is the volume of samples (mL), and *m* is the initial mass of oat flour and OS in the reaction solution (mg).

### 2.7. Monosaccharide Composition

Samples at a concentration of 5 mg/mL were dissolved in ultrapure water and boiled for 60 min. Each sample was filtered through a 0.22 μm water-based filter. Saccharide profiles were analyzed using high-pressure ion chromatography (HPIC) (DIONEX ICS-500+SP-5, Thermo Fisher Scientific, Waltham, MA, USA) equipment fitted with a Dionex CarboPac PA200 column. A mixture of malto-oligosaccharides (G1–7) at a concentration of 50 µg/mL was utilized to generate a standard curve for saccharide quantification. The percentages of glucose (G1), maltose (G2), maltotriose (G3), maltotetraose (G4), maltopentose (G5), and maltohexaose (G6) were determined by calculating the respective peak integrations using the equipment software [21].

### 2.8. Molecular Weight Distribution

The molecular weight distributions of OS and EHOS were obtained by modifying a method from a previous study [22]. The samples were dissolved (5 mg/mL) in ultrapure water by boiling for 60 min. Each sample was filtered using a 0.45 μm water-based filter. The molecular weight distribution was determined using a high-performance liquid chromatography (HPLC) system (Waters 1525EF, Waters, Milford, CT, USA) equipped with an UltrahydrogelTM Linear 300 mm × 7.8 mm column equilibrated at 40 °C. The flow rate of the mobile phase (0.1 M NaNO_3_) was 0.5 mL/min. Glucose (Mw 180), Dextran T-5 (Mw 2700), Dextran T-10 (Mw 9750), Dextran T-150 (Mw 135,030), Dextran T-300 (Mw 300,600), and Dextran T-2000 (Mw 2,000,000) were used as standard samples. The calculation of the polydispersity index (PDI) involves obtaining the quotient of the weight-average molecular weight (Mw) and the number-average molecular weight (Mn). Mathematically, it is expressed in the form PDI = Mw/Mn.

### 2.9. Side Chain Distribution

According to a previous study [23], samples were suspended in 75% ethanol, and the precipitate was collected by centrifuging at 7000× *g* for 5 min. The precipitates were dried in an oven after repeating the above steps. 50 mg of samples were moistened with 0.5 mL ultrapure water and dissolved in 4.5 mL DMSO. The resulting suspension was gelatinized in a boiling water bath for 30 min and then transferred into an enzyme reactor at 25 °C overnight. Afterward, 1 mL of the solution was transferred using a pipette into 6 mL of anhydrous ethanol. The resulting mixture was centrifuged at 2800× *g* for 20 min to separate and discard the supernatant. After the ethanol evaporated completely, 10 mL of boiled ultrapure water was added to the precipitate and boiled for 10 min before immediate transfer into a water bath at 50 °C. After the temperature became constant, 40 µL of Gly-HCl (3.7535 mg/mL, pH 3.5) was added, and the mixture was treated with isoamylase (2 µL) for 48 h. The enzymatic reaction was terminated by placing the sample in a boiling water bath. Each sample was filtered through a 0.22 µm water-based filter, and then 20 µL was injected into a high-performance size-exclusion chromatography (HPSEC) (Alliance E2695, Waters, Milford, CT, USA) system equipped with a tandem column (Shodex SB-804 HQ, SB-802.5 HQ) and a guard column (Shodex SB-G). Ultrapure water was used as the mobile phase, with a flow rate of 1 mL/min. The column temperature was set at 80 °C, and the detector temperature was set at 50 °C. Aliquots of pullulanase (1 mg/mL) of different molecular weights (M-7, P-5, P-10, P-20, P-50, and P-100) were used as standard samples.

### 2.10. Rheological Measurement

#### 2.10.1. Frequency Sweep Test

The OS or EHOS was transferred onto the plane of the rheometer at 25 °C. The samples were allowed to equilibrate for approximately 5 min before testing. The rheological properties of the samples were measured using a DHR-2 rheometer (Discovery Hybrid Rheometer; TA Instrument, New Castle, DE, USA) equipped with a 40 mm diameter parallel plate [24]. A gap of 1.0 mm was employed to conduct the dynamic rheological test at 25 °C. For dynamic oscillatory tests, the storage modulus (G′) and loss modulus (G″) were measured in frequency sweep mode. The strain was fixed at 1% in the linear viscoelastic region, and the oscillatory frequency ranged from 0.1 to 100 Hz.

#### 2.10.2. Steady Shear Test

In the shear rate sweep mode, both OS and EHOS underwent shearing within a shear rate range from 0.1 to 100 s^−1^ to characterize the flow behavior. All samples were prepared following the exact same procedure described above.

#### 2.10.3. Time Sweep Test

After adding the mixed enzymes, the OS pastes were transferred onto the plane of the rheometer at 60 °C. Time sweep measurement of the static rheological behaviors of samples simulated the process of the enzymatic hydrolysis reaction [25]. The strain was fixed at 1% in the linear viscoelastic region, and the oscillatory frequency and time were 1.0 Hz and 60 min, respectively.

#### 2.10.4. Peak Hold Step Test

After adding the mixed enzymes, the OS pastes were transferred onto the plane of the rheometer at 60 °C. A peak hold step test was carried out to mimic the enzymatic hydrolysis process. To simulate this process, a constant shear rate was applied. The step was shearing at 1.0 s^−1^ for 3600 s. To explore the influence of different ratios of pullulanase and isoamylase on the rheological behavior of OS during the enzymatic process, the test was conducted on OS with pullulanase and isoamylase added in the ratios of 2:1, 1:1, and 1:2.

### 2.11. Characterization of Emulsion

#### 2.11.1. Emulsion Activity Index (*EAI)* and Emulsion Stability Index (*ESI*)

The sample solutions (1%, *w*/*v*) were prepared by dissolving OS/EHOS in distilled water under stirring overnight to achieve hydration. The emulsions were manufactured by mixing solutions with rapeseed oil at an oil/water volume ratio of 1:3, and then homogenized with a T18 homogenizer for 2 min at 12,000 rpm [26]. Immediately after homogenization was completed (at 0 min), and 10 min later, the prepared emulsions were added to 0.1% sodium dodecyl sulfate (SDS) to create a solution diluted 100-fold [27]. Emulsion absorbance was measured with a T6 UV-Vis spectrometer (PERSEE, China) at 500 nm. The *EAI* and *ESI* of OS and EHOS were determined according to Formulas (3) and (4), respectively.(3)$$EAI\;(m^{2}/g)=(2.303\times 2\times N\times A_{0})/10,000\times \rho \times L\times C$$(4)$$ESI\;(\min)=(A_{0}\times 10)/(A_{0}-A_{10})$$
where *A*_0_ and *A*_10_ represent the absorbance of the diluted emulsion after 0 and 10 min of homogenization, respectively, *N* stands for the dilution multiple (100), *ρ* is the volume fraction of rapeseed oil (0.25), *L* is the path length of the cuvette (1 cm), and *C* represents the concentration of OS or EHOS (g/mL).

#### 2.11.2. Z-Average Size and Zeta Potential Measurement

The emulsions were prepared by mixing solutions (1%, *w*/*v*) with 3% rapeseed oil and then homogenized with a T18 homogenizer for 2 min at 12,000 rpm. The Z-average size and zeta potential of emulsions were determined by a Zeta Sizer. Prior to measurement, the samples were diluted 10 times with distilled water.

#### 2.11.3. Storage Stability Determination

The emulsions mentioned above were transferred into transparent glass bottles. Then, a camera was used to record the storage while the emulsions were stored at 4 °C. Emulsions were photographed at 0, 1 h, and 24 h to assess their storage stability.

### 2.12. Statistical Analysis

The data were expressed as mean ± standard deviation (SD). SPSS 22.0 software was used to conduct a significance analysis of the data. Means were compared using one-way analysis of variance (ANOVA), followed by Duncan’s multiple comparing tests. Letters represent significant differences (*p* < 0.05).

## 3. Results

### 3.1. Stability Analysis

Oat-based beverages are complex systems, rich in carbohydrates, proteins, and fats, and solving the stability problem of starch is an important prerequisite to improving its quality [28]. First of all, we investigated the effect of amylase and debranching enzyme addition on the stability of oat milk. Figure 1a illustrates that the stability of oat milk improved with the increase in α-amylase addition and a maximum value obtained at 1.00%, beyond which the stability decreased. In addition, stability was enhanced by the addition of amyloglucosidase, with the highest stability observed at 0.2% (Figure 1b). Increasing the amount of amyloglucosidase within a certain range was beneficial in improving the stability of oat milk, although it had no positive effect if excessive; this is consistent with the trend of Junejo et al. [29]. The stability increased and then decreased as the content of debranching enzyme increased from 0.15% to 0.35%, and the maximum value was obtained at 0.25% (Figure 1c). As the debranching mechanisms are different—pullulanase has a slow outward action, while isoamylase hydrolyzes both the internal and external branches of amylopectin—we studied the effect of the ratio of pullulanase to isoamylase on oat milk stability [30]. As shown in Figure 1d, stability increased with an increase in the amount of isoamylase, and was significantly improved between 55.51% and 66.68% when the amount of addition was changed from 2:1 to 1:2. Apparently, the ratio of pullulanase to isoamylase was the key variable in improving the stability of oat milk (Figure 1d). In summary, the optimal enzymatic process for oat milk was determined to be 1% α-amylase, 0.2% amyloglucosidase, 0.25% debranching enzyme, and a 1:2 ratio of pullulanase to isoamylase. To further understand the effect of debranching enzyme ratio on oat milk stability, we measured starch hydrolysis degree during a 60 min enzyme treatment. As shown in Appendix A, the trends of oat milk and OS hydrolysis degree curves were similar. After 60 min of reaction, the hydrolysis degree of oat milk reached a maximum of 48.54%, whereas that of oat starch reached 61.44%. In addition, after the same reaction time, the hydrolysis efficiency of the sample with the pullulanase and isoamylase ratio of 1:2 was significantly higher than that of the other two groups. This was consistent with the oat milk stability results, which showed that the higher the degree of hydrolysis, the more stable the oat milk [10]. According to the hydrolysis degree of the oat milk, three reaction times (2 min, 25 min, and 40 min) were chosen as the initial, middle, and final reaction stages, which corresponded to the hydrolysis degree of OS.

### 3.2. Characterization of OS and EHOS

Table 1 shows the changes in the particle size and zeta potential of OS and EHOS with different enzyme proportions and at various time points. The size decreased with increasing enzymatic hydrolysis degree, which was in line with the results from oat starch enzymatically hydrolyzed by α-amylase and pullulanase [31]. Compared to the OS, which had an average size of 1990.33 ± 9.02 nm, the size of the EHOS with more isoamylase was smaller: the granule size of the minimum sample was reduced by a factor of almost 6 (330.93 ± 2.72 nm). The reduction in granule size of hydrolysates could be attributed to the cleavage of starch chains being induced during the hydrolysis. In addition, zeta potential is one of the essential factors known to affect stability. The absolute zeta potential values of EHOS increased with increasing enzymatic hydrolysis degree and isoamylase amount, suggesting improved colloidal stability. The 40PI1:2 exhibited the largest absolute zeta potential value, which might be associated with the increase in hydrolysis. In addition, the decrease in particle size improved the stability of dispersion, and the overall net electrostatic repulsion between the particles increased [32].

The amylose contents of OS and EHOS are listed in Table 1. The amylose content of OS was 27.68 ± 0.69%, a value in the range of 25.2% to 29.4% [33]. After synergistic enzymatic hydrolysis, the amylose content dropped notably. In the final reaction stage, as the pullulanase-to-isoamylase ratio shifted from 2:1 to 1:2, the amylose content decreased from 15.33 ± 0.78% to 4.69 ± 0.40%. This decrease was due to the synergistic action of amylase and the debranching enzymes, which cleaved both the α-1,6-glycosidic and α-1,4-glycosidic bonds of starch and produced short linear chains. Furthermore, it was demonstrated that an optimal yield of linear chains with smaller molecular weights could be attained at a 1:2 ratio of pullulanase to isoamylase.

### 3.3. Structural Characteristics

#### 3.3.1. Saccharide Composition Analysis

To further investigate the effect of synergistic amylase and debranching enzyme catalysis on the fine structure of oat starch, the saccharide composition of the hydrolysates was determined using the HPIC system. As shown in Figure 2, the content of G1 and G2 in the hydrolysate increased significantly with the increase in reaction time. This resulted in their gradual emergence as the predominant products in the hydrolysate, a finding that was consistent with the results of the previous study [34]. In addition, G3~G6 increased and then decreased; the increase was due to long chain splitting, while the subsequent decrease was because G3~G6 was further utilized and decomposed by α-amylase and amyloglucosidase as the reaction time lengthened. The preliminary product of the enzymatic hydrolysis of starch was designated G6. The 2PI2:1 sample exhibited the highest content of G6, which may be attributed to the diminished hydrolysis efficiency of the 2:1 group relative to the 1:2 group [35]. The 40PI1:2 sample contained the largest proportion of small molecule glucose, so by reducing the particle size, it is possible to delay gravitationally induced segregation and improve the stability of oat milk [36]. However, as the glucose content increased, the flavor of the oat milk improved. Moreover, glucose as a basic energy source gives oat milk its potential to provide energy [37].

#### 3.3.2. Molecular Weight Distribution Analysis

The HPSEC system was employed to observe variations in the molecular weight size distribution of the hydrolysates throughout the different treatments. Each curve was divided into high molecular weight (HMW, peak 1), intermediate molecular weight (IMW, peak 2 and/or 3), and low molecular weight (LMW, peak 4) [38]. The values of weight-average molecular weight (Mw) and polydispersity index (PDI) are listed in Appendix A. As the degree of hydrolysis deepened, the values of Mw and PDI were reduced to 299732 Da and 1.15 from the initial 1,078,797 Da and 13.65, indicating that the original macromolecules in soluble starch degraded rapidly and the polydispersity in the system was reduced [22]. This finding aligns with the fact that particle size diminished as the degree of hydrolysis rose. Based on the Mw values of amylose chains and soluble amylopectin, it was deduced that the main OS peak 1 transformed into peak 2 and peak 3 [39]. As illustrated in Figure 3, the proportions of peak 2 and peak 3 decreased significantly. These reductions can be attributed to the synergistic action of amylase and debranching enzymes leading to the degradation of amylose, converting the long chains into relatively short linear chains, and releasing glucose molecules [40]. The enhanced response signal at peak 4 was due to internal and external cleavage by isoamylase, where the molecular weight distribution became narrower and more homogeneous, and the proportion of small molecules of glucose in the hydrolysate increased.

#### 3.3.3. Chain Length Distribution Analysis

Drawing on previous research, the chain length distribution of amylopectin was categorized into three regions according to the degree of polymerization (DP) range: long-branch chains (LBC, DP > 37), intermediate-branch chains (IBC, DP 13~36), and short-branch chains (SBC, DP ≤ 12) [41]. The debranching enzyme specifically cleaves the α-1,6 glycosidic bonds, creating a looser internal structure and yielding hydrolysates with short linear glucans [42]. As seen in Appendix A, the percentage of LBC declined as the enzymatic time progressed, indicating that the content of amylose was continuously decreasing, which is in line with the results of decreasing the amylose content in EHOS. Moreover, the proportion of IBC did not increase significantly when the enzymatic time was extended from 2 min to 25 min (Appendix A). In comparison to Appendix A, a portion of its LBC was transferred to the IBC in a relatively short time. As summarized in Table 2, after enzymatic hydrolysis, the percentage of LBC in the hydrolysate decreased significantly, but the percentage of IBC and SBC increased. Among them, the LBC of 40PI1:2 decreased to 66.52 ± 0.61%, and the proportion of IBC increased to 31.7 ± 0.65%. The debranching process induced oat starch to be subjected to breakage due to the action of enzymes and, as a result, exhibited a reduction in particle size and molecular weight.

### 3.4. Rheological Analysis

The rheological properties of starch hydrolysates are an indication of the quality of starch used in food processing as it relates to the sensory attributes and appearance of the final product. Dynamic rheological properties can characterize the changes in viscoelasticity of hydrolysates obtained from starch at different enzymatic ratios and reaction times. As shown in Figure 4a–c, within the frequency range from 0.1 to 100 Hz, the storage modulus (G′) was always higher than the loss modulus (G″), indicating the solid-like state of starch hydrolysates. Compared to OS, the G′ and G″ of EHOS both witnessed a decrease whose extent grew gradually, along with the progress of the enzymatic hydrolysis, as both short chain amylose and small molecule sugar accounted for low viscous and elastic properties [43]. This is in line with the findings of Li, Wang, Lee, and Li [8] on the hydrolysis of oat starch by adding different amounts of pullulanase. As anticipated in Figure 4d–f, a shear-thinning behavior can be observed clearly as the shear rate increased and the viscosity of the samples declined with the increase in the degree of hydrolysis. In addition, when the ratio of pullulanase to isoamylase was 1:2, the starch hydrolysates showed the smallest viscosity at all time points, which may result from the largest hydrolysis degree (Appendix A). As the degree of enzymatic hydrolysis increased, the hydrolysate pastes formed looser structures than those of OS; this can be ascribed to the decrease in molecular weight and the unravelling and alignment of the polymer molecules.

In order to further investigate the effect of different ratios of pullulanase and isoamylase on the rheological property of OS during the enzymatic hydrolysis process, time sweep and peak hold tests were carried out. Time sweep permits the determination of changes in the elasticity and viscosity of hydrolyzed pastes during enzymatic hydrolysis with different enzyme ratios. Figure 5a,b shows the G′ and G″ of the hydrolysate pastes obtained from different ratios of pullulanase and isoamylase during 60 min of enzymatic hydrolysis. The modulus of elasticity and the viscosity of OS remained constant throughout the course of the experiment, exhibiting an elastic behavior (G′ > G″) [44]. As shown in Figure 5a, there was no significant change in G′ in the early stage (0–2 min) of the reaction, as the ratio of pullulanase to isoamylase changed from 2:1 to 1:1. However, at the later stage (time > 25 min) of the reaction, there was an increase in the rate of reduction of G′. As the ratio changed to 1:2, the value of G′ decreased significantly, even in the early stages of enzymatic digestion. Likewise, the G″ values of all samples showed a similar trend. The apparent viscosity shown in Figure 5c decreased rapidly at the beginning and then levelled off, which was in agreement with the decrease in particle size and amylose content (Table 1) because EHOS, with a smaller molecular size or degree of polymerization, exhibits smaller viscosity [45]. Evidently, hydrolysis significantly reduced the strength of the starch gels and weakened the intermolecular interactions, which is consistent with static rheological properties (Figure 4). It was also consistent with the trend of viscoelastic analysis of the hydrolysates in oat milk (Appendix A). The above results suggest that the proportion of debranched enzyme plays an important role in designing the fluid properties of starch hydrolysates, which undoubtedly affects the quality of the related product.

### 3.5. Characterization of the Emulsion Stability

The emulsifying property of the biopolymers in the beverage plays a vital role in its stability because fat plays an important part in providing nutrition and flavor [46]. We then evaluated the emulsion activity index (*EAI*) and emulsion stability index (*ESI*) of OS and EHOS. As shown in Figure 6, both the EAI and the *ESI* of the starch were enhanced after the enzyme treatment, especially when the ratio of pullulanase to isoamylase was 1:2, when the *EAI* and *ESI* increased from 1.58 m2/g and 39.55 min to 3.67 m2/g and 135.83 min, respectively. The improved emulsifying property of EHOS could be explained by the fact that the synergistic amylase and debranching enzyme catalysis decreased the molecular weight of the starch, which may increase its amphiphilicity. Guo et al. [47] also reported that the emulsifying property of the supernatant starch fraction could be improved by either a higher amylose content or a smaller molecular weight.

We then prepared emulsions using 1% (*w*/*v*) OS and EHOS with 3% rapeseed oil. As presented in Figure 7a, the droplet size of the emulsions made from OS decreased from 3249 nm to 650.07 nm after the synergistic amylase and debranching enzyme treatment, suggesting an increased emulsifying stability of the EHOS, which was in line with the results from *ESI* (Figure 6) [48]. In addition, Figure 7b shows that the zeta potential of the emulsions prepared by OS decreased from −9.36 mV to −51.41 mV after the synergistic amylase and debranching enzyme treatment. Moreover, the absolute value of the zeta potential increased as the ratio of pullulanase to isoamylase was changed to 1:2, indicating that enzyme modification in this ratio increased the electrostatic repulsion of the droplets and could improve the stability of the emulsified system [49]. This is consistent with the above trends in the Z-average size and zeta potential of hydrolysates (Table 1).

Pictures of the emulsions prepared at 0 h, 1 h, and 24 h are shown in Figure 7c. There was obvious delamination in the emulsion prepared with OS at 1 h, indicating its poor emulsifying ability. In contrast, the emulsion prepared with hydrolysates showed no significant change after 1 h of storage, indicating that hydrolysis improved the emulsifying stability of the starch. After 24 h, the emulsions prepared with hydrolysates in the ratios of 2:1 and 1:1 of pullulanase and isoamylase both showed separation, whereas the emulsion prepared with 1:2 showed only slight phase separation, indicating that it maintained good emulsion stability. It was also demonstrated that the emulsion at a debranching enzyme ratio of 1:2 was due to its minimal molecular weight, significant breakage of the long chains, and the production of the largest number of small glucose molecules.

## 4. Conclusions

This study first revealed the stabilization mechanism of oat milk under the combined action of amylase and debranching enzymes. The impact of oat starch following enzymatic hydrolysis on the stability of oat milk was also examined. The results of the fine structure tests of the hydrolysates demonstrated that the starch was fragmented into relatively short branched chains, and the highest concentration of glucose was released when the ratio of pullulanase to isoamylase was 1:2 in comparison to the OS. Additionally, the G′ and viscosity were at their lowest as the ratio of pullulanase to isoamylase was 1:2, and the emulsion stability index of the starch hydrolysates was enhanced from 39.55 min to 135.83 min. In conclusion, the modified structure and improved emulsification properties of oat starch synergistically hydrolyzed by amylase and debranching enzyme show its potential for a wide range of applications in the food industry, as well as providing new perspectives for improving the stability of oat milk. This includes conducting large-scale trials to evaluate process optimization under production conditions and developing control strategies for texture variations during long-term storage.

## Figures and Tables

**Figure 1 foods-14-01271-f001:**
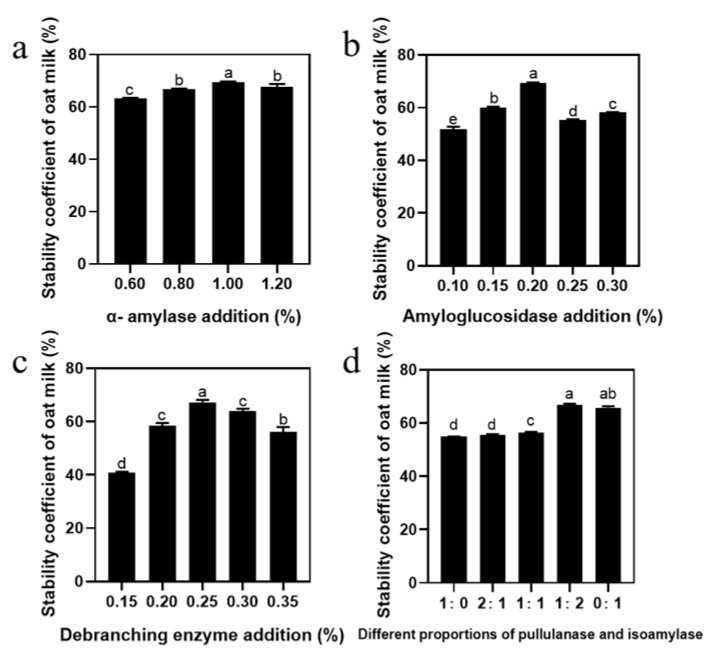
Single-factor test analysis of the stability of oat milk: effect on the stability of oat milk of (**a**) α-amylase, (**b**) amyloglucosidase, (**c**) debranching enzyme, and (**d**) different proportions of pullulanase and isoamylase addition. Data are shown as mean ± standard error. Letters represent significant differences (*p* < 0.05).

**Figure 2 foods-14-01271-f002:**
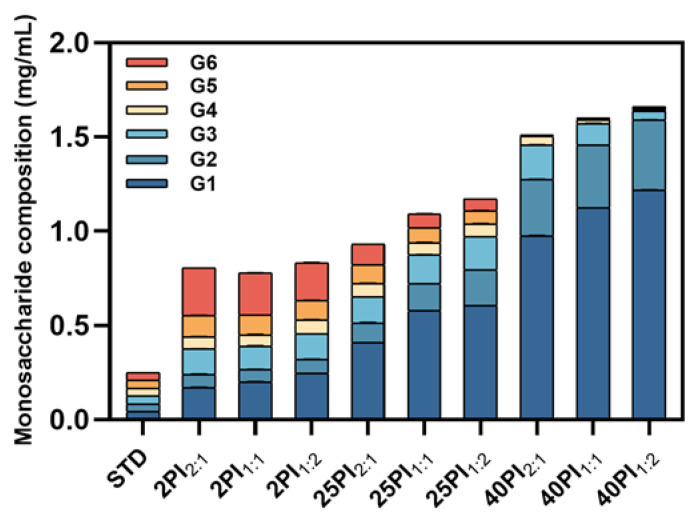
The monosaccharide content of EHOS. OS, oat starch; 2PI_2:1_/2PI_1:1_/2PI_1:2_, treated for 2 min at pullulanase to isoamylase ratios of 2:1, 1:1, and 1:2; 25PI_2:1_/25PI_1:1_/25PI_1:2_, treated for 25 min at pullulanase to isoamylase ratios of 2:1, 1:1, and 1:2; 40PI_2:1_/40PI_1:1_/40PI_1:2_, treated for 40 min at pullulanase to isoamylase ratios of 2:1, 1:1, and 1:2. G1, glucose; G2, maltose; G3, maltotriose; G4, maltotetraose; G5, maltopentose; G6, maltohexaose.

**Figure 3 foods-14-01271-f003:**
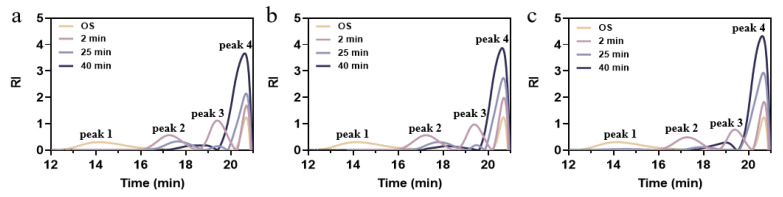
Molecular weight distribution of OS and EHOS with (**a**) 2:1, (**b**) 1:1, and (**c**) 1:2 ratios of pullulanase and isoamylase. OS, oat starch; 2 min, treated for 2 min; 25 min, treated for 25 min; 40 min, treated for 40 min.

**Figure 4 foods-14-01271-f004:**
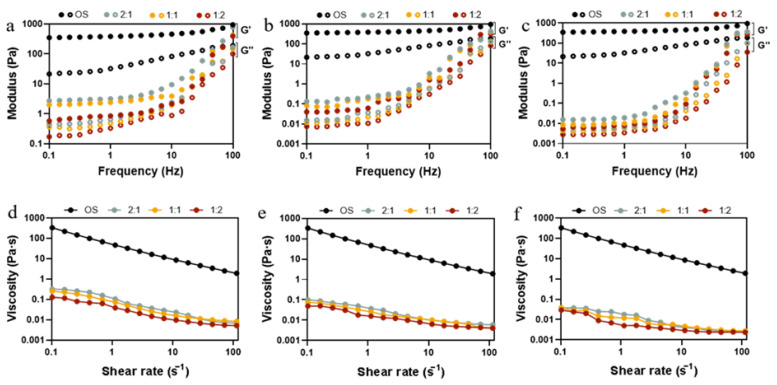
Variation of storage modulus (G′, filled symbols) and loss modulus (G″, empty symbols) with frequency (Hz) and flow behavior for OS and EHOS at reaction times of (**a**,**d**) 2 min, (**b**,**e**) 25 min, and (**c**,**f**) 40 min. 2:1, treated with pullulanase and isoamylase at a ratio of 2:1. 1:1, treated with pullulanase and isoamylase at a ratio of 1:1. 1:2, treated with pullulanase and isoamylase at a ratio of 1:2.

**Figure 5 foods-14-01271-f005:**
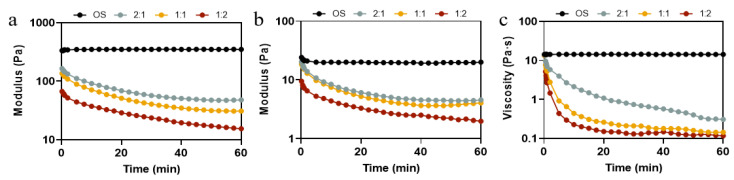
(**a**) G′ and (**b**) G″ curves of OS treated by different ratios of debranching enzyme, and (**c**) viscosity curve of OS treated by different ratios of debranching enzyme. 2:1, treated with pullulanase and isoamylase at a ratio of 2:1. 1:1, treated with pullulanase and isoamylase at a ratio of 1:1. 1:2, treated with pullulanase and isoamylase at a ratio of 1:2.

**Figure 6 foods-14-01271-f006:**
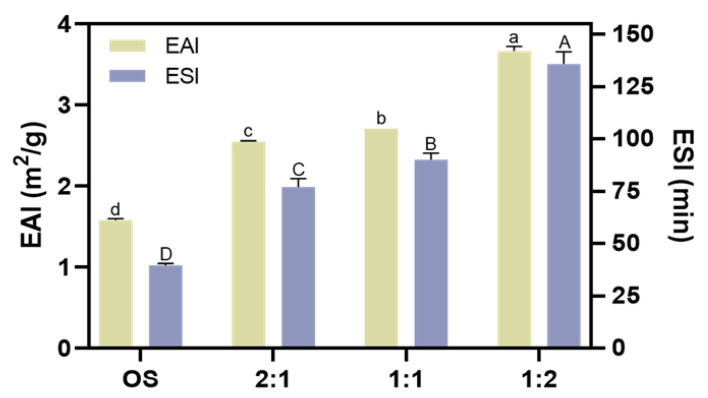
Effect of different ratios of debranching enzymes on the emulsifying activity index (*EAI*) and emulsifying stability index (ESI). Data are shown as mean ± standard error. Letters represent significant differences (*p* < 0.05). 2:1, treated with pullulanase and isoamylase at a ratio of 2:1. 1:1, treated with pullulanase and isoamylase at a ratio of 1:1. 1:2, treated with pullulanase and isoamylase at a ratio of 1:2.

**Figure 7 foods-14-01271-f007:**
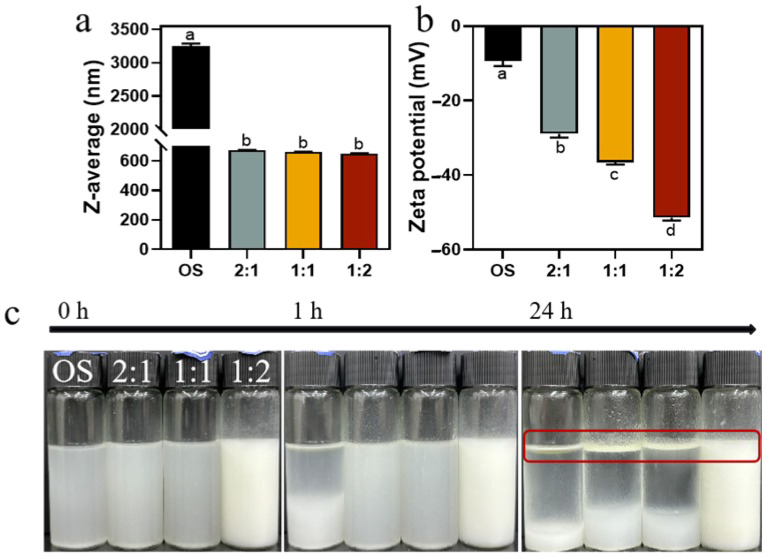
(**a**) Z-average size and (**b**) zeta potential of emulsions prepared using OS and EMOS. (**c**) Visual appearance of emulsions stabilized by OS and EMOS after storing for 0 h, 1 h, and 24 h. Data are shown as mean ± standard error. Letters represent significant differences (*p* < 0.05). 2:1, treated with pullulanase and isoamylase at a ratio of 2:1. 1:1, treated with pullulanase and isoamylase at a ratio of 1:1. 1:2, treated with pullulanase and isoamylase at a ratio of 1:2.

**Table 1 foods-14-01271-t001:** Z-average size, zeta potential, and amylose content of OS and EHOS.

	OS	2PI_2:1_	2PI_1:1_	2PI_1:2_	25PI_2:1_	25PI_1:1_	25PI_1:2_	40PI_2:1_	40PI_1:1_	40PI_1:2_
Z-average (nm)	1990.33 ± 9.02 ^a^	677.97 ± 5.17 ^b^	661.10 ± 6.42 ^c^	421.07 ± 2.18 ^f^	536.00 ± 0.69 ^d^	404.27 ± 1.85 ^g^	348.23 ± 1.55 ^i^	477.53 ± 4.45 ^e^	379.97 ± 1.76 ^h^	330.93 ± 2.72 ^j^
zeta potential (mV)	−2.13 ± 0.04 ^a^	−21.60 ± 1.00 ^b^	−26.25 ± 1.08 ^c^	−29.23 ± 0.35 ^d^	−28.61 ± 0.77 ^d^	−30.48 ± 0.17 ^e^	−32.20 ± 0.95 ^e^	−29.59 ± 0.30 ^d^	−29.59 ± 1.99 ^d^	−32.42 ± 0.44 ^e^
amylose content (%)	27.68 ± 0.69 ^a^	25.78 ± 1.36 ^b^	13.29 ± 0.32 ^e^	10.05 ± 0.95 ^g^	21.05 ± 0.79 ^c^	11.60 ± 0.52 ^f^	10.16 ± 0.27 ^g^	15.33 ± 0.78 ^d^	9.06 ± 0.49 ^g^	4.69 ± 0.40 ^h^

Data are shown as mean ± standard error. Letters represent significant differences (*p* < 0.05). OS, oat starch; 2PI_2:1_/2PI_1:1_/2PI_1:2_, treated for 2 min at pullulanase to isoamylase ratios of 2:1, 1:1, and 1:2; 25PI_2:1_/25PI_1:1_/25PI_1:2_, treated for 25 min at pullulanase to isoamylase ratios of 2:1, 1:1, and 1:2; 40PI_2:1_/40PI_1:1_/40PI_1:2_, treated for 40 min at pullulanase to isoamylase ratios of 2:1, 1:1, and 1:2.

**Table 2 foods-14-01271-t002:** Percentage of side chains in debranched OS and EHOS.

Sample	Side Chain Distribution (%)			
	DP < 12	DP13–24	DP25–36	DP ≥ 37
OS	0.51 ± 0.03 ^g^	0.63 ± 0.03 ^i^	1.38 ± 0.06 ^i^	97.48 ± 0.08 ^a^
2PI_2:1_	0.28 ± 0.02 ^h^	0.94 ± 0.02 ^h^	2.05 ± 0.08 ^h^	96.66 ± 0.08 ^a^
2PI_1:1_	0.73 ± 0.06 ^f^	1.92 ± 0.03 ^f^	2.61 ± 0.21 ^g^	94.68 ± 0.25 ^b^
2PI_1:2_	1.00 ± 0.06 ^e^	4.29 ± 0.24 ^e^	5.61 ± 0.21 ^e^	89.03 ± 0.50 ^d^
25PI_2:1_	0.75 ± 0.03 ^f^	1.42 ± 0.05 ^g^	2.70 ± 0.11 ^g^	95.06 ± 0.07 ^b^
25PI_1:1_	0.97 ± 0.01 ^e^	1.96 ± 0.07 ^f^	4.40 ± 0.03 ^f^	92.59 ± 0.03 ^c^
25PI_1:2_	1.61 ± 0.05 ^b^	7.53 ± 0.07 ^d^	9.00 ± 0.15 ^d^	81.79 ± 0.17 ^e^
40PI_2:1_	1.08 ± 0.01 ^d^	8.04 ± 0.01 ^c^	15.97 ± 0.52 ^c^	82.56 ± 2.19 ^e^
40PI_1:1_	1.18 ± 0.02 ^c^	10.08 ± 0.02 ^b^	18.19 ± 0.22 ^b^	70.49 ± 0.25 ^f^
40PI_1:2_	1.71 ± 0.01 ^a^	12.44 ± 0.37 ^a^	19.26 ± 0.28 ^a^	66.52 ± 0.61 ^g^

Data are shown as mean ± standard error. Letters represent significant differences (*p* < 0.05). OS, oat starch; 2PI_2:1_/2PI_1:1_/2PI_1:2_, treated for 2 min at pullulanase to isoamylase ratios of 2:1, 1:1, and 1:2; 25PI_2:1_/25PI_1:1_/25PI_1:2_, treated for 25 min at pullulanase to isoamylase ratios of 2:1, 1:1, and 1:2; 40PI_2:1_/40PI_1:1_/40PI_1:2_, treated for 40 min at pullulanase to isoamylase ratios of 2:1, 1:1, and 1:2. DP < 12, the degree of polymerization is less than 12; DP13–24, the degree of polymerization ranges from 13 to 24; DP25–36, the degree of polymerization (DP) ranges from 25 to 36; DP ≥ 37, the degree of polymerization is greater than 37.

## Data Availability

The data presented in this study are available on request from the corresponding author due to privacy.

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
