# Peer review of "Synergistic Amylase and Debranching Enzyme Catalysis to Improve the Stability of Oat Milk"

_foods, 2025, doi:10.3390/foods14071271_

Round 1

Reviewer 1 Report

Comments and Suggestions for Authors

The manuscript entitled “Synergistic amylase and debranching enzyme catalysis to improve the stability of oat milk” deals with the effects of different enzyme combinations on the stability of oat milk and the properties of starch in oats by the addition of α-amylase, amyloglucosidase, pullulanase and iso-amylase. The authors have shown that the molecular weight, amylose content, and side chain length distribution of the starch decreased and the rheological and emulsifying properties of the starch hydrolysates were improved. With these results they have demonstrated that debranching enzymes enhanced the enzymatic hydrolysis of beverages and improved the physicochemical properties and stability of oat milk. These results are interesting because they show technological advances for oat milk beverages. The manuscript needs English revision, but the conclusions are supported by data. Some points should be addressed to improve the clarity for the reader.

Materials and Methods

2.2

“The effects of α-amylase addition on the stability of oat milk were investigated by adding 0.20% (w/w, on starch dry basis) of amyloglucosidase, 0.22% of debranching enzyme (pullulanase: isoamylase, 1:1), 0.60%, 0.80%, 1.00% and 1.20% of α-amylase, to the slurry, respectively, and...”

This description is confusing. The enzymes were added to the slurry? Which slurry? In wich step of oat milk preparation? “respectively” to what? Please clarify. Please do so also for 2.2.2, 2.2.3, and 2.2.4.

2.3

Why the stability coefficient is defined as “R(%),” and why is this letter not used in the result section?

“2.5 Preparation of enzyme hydrolyzed oat starch (EHOS)” which “sample suspensions”?

“After cooling to 60 °C, addition of various enzymes according to the optimal enzymatic process, the debranching enzyme with proportions of 2:1, 1:1 and 1:2 (w/w, on starch dry basis) were added to the OS pastes, respectively, to promote enzymatic hydrolysis reaction for 2 min, 25 min and 40 min, respectively.” Rewrite sentence. It is confusing.

“2.8. Molecular weight distribution” molecular weight distribution of what?

“2.10 Rheological measurement” what “pastes” mean? Please revise all these measurement descriptions. Make sure to inform what is being measured and which sample.

2.12: what “SPSS” means?

Results

3.1

“In summary, the optimal enzymatic process for oat milk was determined to be 1% α-amylase, 0.2% amyloglucosidase, 0.25% debranching enzyme and 1:2 ratio of pullulanase to isoamylase.” All together? But the study was performed with each one independently? Or all the enzymes were used in all the tests? If so, this information needs to be in the figure legend.

“three time points” instead of “points” it would be better to say reaction time, period of time, process time. “point” is not a parameter.

Table 1: make sure that all tables and figures can stand alone without referring to text. So, please, inform all the abbreviations in the footnote.

Figure 2: the same. What is G1, G2,...?

Does the increase in glucose content of the samples have an impact on the nutritional properties of the beverage? Even if the goal of the paper was not to access this feature, some comments about that, with literature backup, must be in the text.

Figure 3: The meaning of 2 min, 25 min, and 40 min should be in the figure legend.

Comments on the Quality of English Language

It needs improvement.

Reviewer 2 Report

Comments and Suggestions for Authors

The introduction provides a well-rounded background on oat milk stability issues and enzymatic hydrolysis but could better highlight the novelty of this study compared to previous research (Page 3, Lines 29-50). The methodology is comprehensive but lacks justification for selecting specific enzyme concentrations and reaction times, which should be supported by references or prior optimization studies (Page 8, Lines 85-100). The results section effectively presents key findings, but some sections, particularly on saccharide composition and molecular weight distribution, are overly descriptive and could be streamlined for clarity (Pages 14-17, Lines 225-310). The statistical analysis is appropriate but would benefit from the inclusion of effect sizes and confidence intervals to strengthen the interpretation of significance (Page 20, Tables 1-3). Figures and tables effectively illustrate the findings, but clearer labeling is needed, especially for rheological and emulsification data (Figures 4-7, Pages 22-26). The discussion integrates findings well with previous literature but lacks a critical evaluation of potential limitations, such as the impact of enzyme-modified oat starch on sensory attributes and consumer acceptance (Page 24, Lines 275-310). The conclusion successfully summarizes the study’s contributions. Still, it should provide clearer recommendations on industrial applications and future research directions, such as potential challenges in large-scale production or storage stability of enzyme-treated oat milk (Page 26, Lines 410-425). While the study presents valuable insights into enzymatic modifications for plant-based beverages, refining the clarity, structure, and practical implications would enhance its impact.

Round 2

Reviewer 2 Report

Comments and Suggestions for Authors

The manuscript has undergone a comprehensive revision, effectively incorporating all suggested modifications to improve clarity, methodological soundness, and scientific relevance. The introduction now articulates the research gap more precisely, and the methodology section includes clear justifications to support reproducibility. The discussion has been enriched with a more in-depth analysis, covering key results, limitations, and comparisons with existing literature. The conclusion highlights the broader significance, potential for scalability, and directions for future research. Terminology has been harmonized, redundancies eliminated, and recent literature added. With these thorough enhancements, the manuscript now meets the standards for publication and represents a coherent, scientifically solid work ready for dissemination.